# Lego Sketch: A Scalable Memory-augmented Neural Network for Sketching Data Streams

Yuan Feng [* 1 2]  Yukun Cao [* 1 2]  Hairu Wang [1 2]  Xike Xie [3 2]  S. Kevin Zhou [3 2]

## Abstract

Sketches, probabilistic structures for estimating item frequencies in infinite data streams with limited space, are widely used across various domains. Recent studies have shifted the focus from handcrafted sketches to neural sketches, leveraging memory-augmented neural networks (MANNs) to enhance the streaming compression capabilities and achieve better space-accuracy trade-offs. However, existing neural sketches struggle to scale across different data domains and space budgets due to inflexible MANN configurations. In this paper, we introduce a scalable MANN architecture that brings to life the *Lego sketch*, a novel sketch with superior scalability and accuracy. Much like assembling creations with modular Lego bricks, the Lego sketch dynamically coordinates multiple memory bricks to adapt to various space budgets and diverse data domains. Our theoretical analysis guarantees its high scalability and provides the first error bound for neural sketch. Furthermore, extensive experimental evaluations demonstrate that the Lego sketch exhibits superior space-accuracy trade-offs, outperforming existing handcrafted and neural sketches. Our code is available at https://github.com/FFY0/LegoSketch_ICML.

## 1. Introduction

The estimation of item frequency in a continuous and never-ending data stream stands as a pivotal task in supporting a broad spectrum of applications in machine learning (Goyal et al., 2012; Aghazadeh et al., 2018; Talukdar & Cohen,

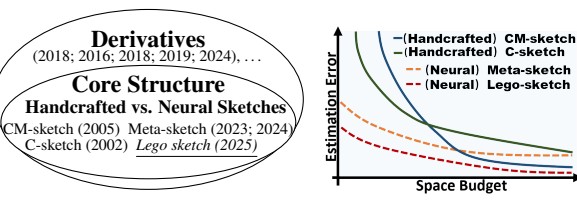

Figure 1: Sketch Literatures

Figure 2: Space-accuracy Trade-off (Aol Dataset)

2014), network measurements (Yu et al., 2013; Yang et al., 2018), and big data analytics (Cormode et al., 2012; Zaharia et al., 2016). Sketches, a typical probabilistic structure, have become essential for representing data streams with sub-linear space and linear time while maintaining accurate item frequency estimates. Two major research direction in sketching techniques have emerged: *handcrafted sketches* (Cormode & Muthukrishnan, 2005; Charikar et al., 2002; Estan & Varghese, 2002; Deng & Rafiei, 2007) and *neural sketches* (Rae et al., 2019; Feng et al., 2024; Cao et al., 2023; 2024).

Handcrafted sketches rely on core structures, such as the CM-sketch (Cormode & Muthukrishnan, 2005) and C-sketch (Charikar et al., 2002), which are comprised of compact 2D arrays, hash functions, and predefined strategies. These handcrafted core structures have since formed the foundation for numerous variants or derivatives (Estan & Varghese, 2002; Deng & Rafiei, 2007), designed to better adapt to the skewed distributions in data streams[4]. In contrast to the decade-old core structures of handcrafted sketches, recent advancements in neural sketches (Cao et al., 2023; 2024; Feng et al., 2024; Rae et al., 2019) have introduced memory-augmented neural networks (MANNs) as a new class of core structures. The MANN-based cores improve sketching performance by leveraging their capability of memory compression and adaptability to distributional patterns of data streams. Figure 1 provides an overview of the literatures on sketches.

Despite recent advancements, existing neural sketches face practical challenges when deployed in real-world settings, particularly in scaling effectively to data streams across diverse domains and varying space budgets. A major scalability issue is that they require retraining when there is a

---

[*]Equal contribution [1]School of Computer Science, University of Science and Technology of China (USTC), China [2]Data Darkness Lab, MIRACLE Center, Suzhou Institute for Advanced Research, USTC, China [3]School of Biomedical Engineering, USTC, China. Correspondence to: Xike Xie <xkxie@ustc.edu.cn>.

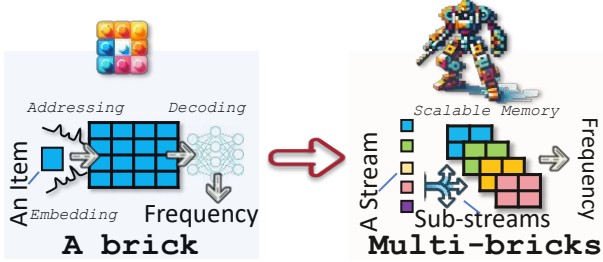

Figure 3: Lego Sketch Overview: The Lego sketch enables a scalable and unified framework capable of adapting to different domains and space budgets, mirroring the modular design seen in dedicate creations built from multiple Lego bricks.

shift in data domains or changes in space budgets. This requirement stems from their reliance on a sample MLP-based embedding module and a fixed-size memory module within the traditional MANN architecture. Also, as demonstrated in Figure 2, the accuracy advantage of existing neural sketches, i.e., meta-sketches, in terms of estimation error over handcrafted sketches, tends to decrease as the space budget increases.

In this paper, we propose a novel neural sketch, called the *Lego sketch*, designed to significantly enhance scalability and improve the space-accuracy trade-offs of existing neural sketches, thereby setting new benchmarks in performance[1]. As depicted in Figure 3, the Lego sketch is initially trained on a single memory brick to acquire basic sketching capabilities across data domains. When dealing with a large-scale data stream, it divides the stream into manageable sub-streams, each controlled by an independent memory brick. The Lego framework is achieved by devising a new MANN architecture from two aspects.

**For scalability**, the Lego sketch overcomes the shortcomings of existing neural sketches under conventional MANN architecture, particularly when used across different data domains and space budgets. First, we propose a novel *normalized multi-hash embedding* technique to enable theoretically provable domain-agnostic scalability (Section 4.1), without necessitating retraining for different data domains. Second, with theoretical foundations (Section 4.2), our approach leverages a *scalable memory* strategy to coordinate multiple memory bricks, avoiding the need for retraining when adapting memory sizes to fit different space budgets.

**For accuracy**, the Lego sketch not only addresses the issue of diminished advantage of existing neural sketches in high budgets but also offers enhancement in accuracy.

---

[1]This paper focuses on the core structure of sketches. Derivatives of a core structure typically incorporates external enhancements, such as filters (discussed in Section 2). Section 3.3 explores derivatives of the Lego sketch, which use it as the core structure, providing a comprehensive analysis.

Specifically, we propose a novel *self-guided weighting loss* (Section 3.2) for dynamically weighting different meta-tasks during self-supervised meta-learning, effectively addressing issues of diminished advantage. Moreover, a pioneering module termed *memory scanning* (Section 3.1), which autonomously reconstructs stream features from the compressed memory, along with other optimization techniques, significantly aids in the estimation, thereby achieving superior space-accuracy trade-offs, as shown in Figure 2.

**Contributions. 1)** We introduce the Lego sketch, a novel neural sketch equipped with a scalable MANN architecture designed for sketching data streams. **2)** The Lego sketch resolves scalability limitations and achieves a superior space-accuracy trade-off through key technical innovations, including *normalized multi-hash embedding*, *scalable memory*, *memory scanning*, and *self-guided weighting loss*. **3)** We also provide theoretical support for the scalability and estimation error analysis, areas previously unexplored in existing neural sketches. **4)** Extensive empirical studies on both real-world and synthetic datasets confirm that the Lego sketch significantly outperforms state-of-the-art methods.

## 2. Related Works

**Sketch** techniques fall into two categories: core sketches and their derivatives, as shown in Figure 1. Core sketches (Cormode & Muthukrishnan, 2005; Charikar et al., 2002; Estan & Varghese, 2002; Deng & Rafiei, 2007; Liu & Xie, 2021; Cao et al., 2023; 2024), also referred to as core structures, provide the fundamental structure for stream processing, where (Estan & Varghese, 2002; Deng & Rafiei, 2007) are variants of (Cormode & Muthukrishnan, 2005; Charikar et al., 2002) under specific conditions. Subsequent derivatives (Zhou et al., 2018; Roy et al., 2016; Yang et al., 2018; Aamand et al., 2024; Hsu et al., 2019; Zhao et al., 2023; Ma et al., 2024; Gao et al., 2024a; Liu & Xie, 2023; Liu et al., 2024; Gao et al., 2024b; Huang et al., 2023) of core structures, depend on the core structure for their foundational performance and incorporate external enhancement with orthogonal add-ons like filters. Notably, recent advances in neural sketches, i.e., the meta-sketch (Cao et al., 2023; 2024), have shown improved accuracy under tight space budgets but face scalability challenges. Our Lego sketch tackles the scalability challenges and optimizes the space-accuracy trade-offs. Moreover, we study how subsequent derivatives can utilize the Lego sketch as their core, detailed in Section 3.3.

**Counter-based summarization** is also a common method for summarizing data streams, identifying frequent items through techniques like MG summary (Misra & Gries, 1982) and SpaceSaving (Metwally et al., 2005). These methods typically excel in insertion-only stream models, with some research also exploring learning-based enhancements (Sha-

Table 1: MANN Architecture in Neural Sketches

| MANN Arch. | Embedding | Memory | Scanning | Loss |
|---|---|---|---|---|
| Meta-Sketch | MLP-based | Fixed | - | Regular Loss $\mathcal{L}_o$ |
| Lego Sketch | Scalable | Scalable | Deepsets-based | Self-guided Loss $\mathcal{L}'$ |

hout & Mitzenmacher, 2024). However, compared to sketch techniques, counter-based methods struggle with dynamic streams that allow deletions or negative weights, while sketch methods naively handle unbounded deletions, making them suitable for broader applications (Cormode & Hadjieleftheriou, 2008; Berinde et al., 2010). Given these differences, counter-based summarization and sketch techniques should not be directly compared; nonetheless, future research could explore the development of an end-to-end neural framework for counter-based summarization.

**Memory-augmented neural networks (MANNs)** (Graves et al., 2016; Danihelka et al., 2016; Weston et al., 2014; Graves et al., 2014) are designed for efficient interaction with external memory through neural networks. Leveraging meta-learning, MANNs facilitate one-shot learning capability of certain tasks across various datasets, avoiding be limited to specific dataset (Santoro et al., 2016; Vinyals et al., 2016; Hospedales et al., 2021). Previous efforts on neural sketches (Cao et al., 2023; 2024) implement this MANN architecture in four functional modules: *Embedding*, *Addressing*, *Memory*, and *Decoding*, where the one-shot learning paradigm is inherited for the single-pass storage procedure in stream processing. As aforementioned, neural sketches struggle with the scalability challenge. Our Lego sketch addresses the challenge by devising a novel MANN architecture. A comparison of our architecture with traditional MANNs is outlined in Table 1.

## 3. Methodology

We consider a standard data stream model to outline the problem of stream item frequency estimation. Consider a data stream $\mathcal{X} = (x_1, ..., x_N)$ consisting of $N$ data items with $n$ distinct elements. Each data item $x_i \in \mathcal{X}$ assumes a value from the item domain set $\mathbb{X} = \{e_1, ..., e_n\}$, where elements in $\mathbb{X}$ are unique. The frequency $f_i$ of element $e_i$ indicates its occurrence in $\mathcal{X}$. Thus, the total frequencies for all $f_i$s add up to $N$, representing the length of the stream. Without causing any ambiguity, we use $x_i$ to refer to either a data item for storing or an element for querying.

### 3.1. Lego Framework

**Overview.** The Lego sketch consists of five modules: *Scalable Embedding*($\mathcal{E}$), *Hash Addressing*($\mathcal{A}$), *Scalable Memory* ($\mathcal{M}$), *Memory Scanning* ($\mathcal{S}$), and *Ensemble Decoding*($\mathcal{D}$). These modules collaborate to provide two types of operations: *Store* and *Query*, controlling the writing and reading of external memory. Each operation is carried out

through a single forward pass of several network modules, ensuring that the computational complexity of a single operation remains constant, independent of the data stream's length. This aligns well with the efficiency requirements for sketching data streams (Charikar et al., 2002; Cormode & Muthukrishnan, 2005; Cao et al., 2023). A high-level overview of these operations is presented in Figure 4.

*Scalable Embedding*($\mathcal{E}$)**.** The primary function of the embedding module is to obtain an embedding vector $v_i$ for a given item $x_i$ during *Store* and *Query* operations. Previous MANN based neural sketches, the meta-sketch (Cao et al., 2023; 2024) and other related works (Rae et al., 2019; Vinyals et al., 2016; Santoro et al., 2016; Feng et al., 2024), leverage conventional encoders like MLPs and CNNs. These encoders embed features of data items within specific domains, aim to extract domain-specific knowledge, such as identifying high- and low-frequency items in specific data stream. However, this approach presents several limitations. As highlighted in the meta-sketch (Cao et al., 2023), dynamic streaming scenarios can lead to robustness challenges, due to shifts between high and low-frequency items. Most importantly, these encoders face domain scalability challenges, which require encoder retraining or even model retraining, while deploying models across diverse data domains. For example, a model trained for web-click stream domain cannot be directly adapted to textual stream domain, letting alone the domain of other modalities. To address these challenges, we propose to adjust the "target" of the embedding module as follows. Without extracting any domain-specific features, we generate embedding vectors $\{v\}$ conforming to a specific skewness[4] range, making the embedding module domain-agnostic. This strategy enhances both the robustness and scalability of models across varied domains.

Inspired by the Hash Embeddings (Tito Svenstrup et al., 2017) in NLP fields, we introduce a novel embedding technique, termed *normalized multi-hash embedding*, as below:

$$v_i = \mathcal{E}(x_i) = \frac{(V_{\mathcal{H}_1(x_i)}, V_{\mathcal{H}_2(x_i)}, ..., V_{\mathcal{H}_{d_1}(x_i)})}{\|(V_{\mathcal{H}_1(x_i)}, V_{\mathcal{H}_2(x_i)}, ..., V_{\mathcal{H}_{d_1}(x_i)})\|_1}$$

Specifically, it comprises a learnable vector $V$ and a set of $d_1$ independent hash functions $\{\mathcal{H}_1, ...\mathcal{H}_{d_1}\}$. Each hash function maps $x_i$ to an index of $V$, retrieving corresponding values from $V$ based on $d_1$ indices, yielding $v_i \in \mathbb{R}^{d_1}$. Finally, $v_i$ undergoes an $L_1$ normalization. The idea lies in the utilization of hash-based random mappings, ensuring domain-agnostic scalability as the analysis in Section 4.1. Furthermore, the $L_1$ normalization effectively maintains the stability of $L_1$ accumulation across different items in additive memory storage, thereby enhancing estimation accuracy[2].

---

[2]Related ablation studies are detailed in Section 5.5.

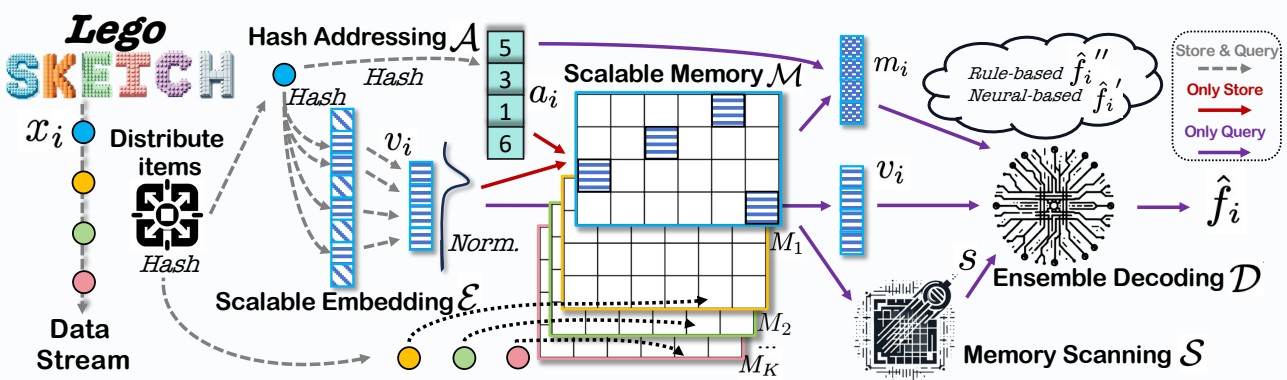

Figure 4: Framework of the Lego Sketch (When an item $x_i$ is stored, it undergoes $\mathcal{E}$ and $\mathcal{A}$, obtaining the embedding vector $v_i$ and address vector $a_i$, which are subsequently stored into the distributed memory brick $M$ within $\mathcal{M}$. When querying an item $x_i$, its embedding vector $v_i$ and address vector $a_i$ are obtained in the same way. These, combined with stream characteristics $s$ reconstructed from the current memory brick by $\mathcal{S}$, are input into $\mathcal{D}$ for frequency estimation $\hat{f}_i$.)

***Hash Addressing*** $(\mathcal{A})$. To align with scalable hash embeddings, our approach simplifies the learning-based addressing in the meta-sketch by employing an additional set of $d_1$ hash functions $\{\mathcal{H}'_i\}_{i \leq d_1}$ for addressing. This method seeks out $d_1$ positions within the range of $[1, d_2]$, subsequently transforming them into a sparse address vector $a_i \in \mathbb{R}^{d_1 \times d_2}$. In the sparse address vector $a_i$, positions mapped by the hash functions are assigned a value of 1, while all other positions retain a value of 0.

$$a_i = \mathcal{A}(x_i) = \text{SparseVector}(\mathcal{H}'_1(x_i), \ldots, \mathcal{H}'_{d_1}(x_i))$$

***Scalable Memory*** $(\mathcal{M})$. With conventional MANNs, the neural structures (Cao et al., 2023; 2024; Rae et al., 2019; Feng et al., 2024) employ a single, fixed-size dense memory block $M \in \mathbb{R}^{d_1 \times d_2}$ to store embedding vectors. Compared to the counter array used in handcrafted sketches, $M$ offers a superior dense compression capability. However, the fixed size of $M$ presents challenges for the scalability in terms of varied space budgets (i.e. the size of $M$): the retraining necessity and increasing cost for training on larger budget. Thus, we introduce the *Scalable Memory*, which offers training-independent memory scalability for the Lego sketch. This module manages $K$ memory bricks, $M_1, \ldots, M_K$, using a hash function to uniformly distribute items in streams across these bricks. This approach allows for memory resizing by simply increasing $K$, thereby eliminating the need for retraining when adjusting the space budget. For example, in Section 5.2, we easily scale the overall memory size up to 140MB for 100 million-level streams.

Specifically, when storing an item $x_i$, it is distributed to a specific memory brick $M_{\mathcal{H}(x_i)}$, by a hash function $\mathcal{H}$. In this way, the original data stream $\mathcal{X}$, is partitioned into $K$ sub-streams $\mathcal{X}'_1, \ldots, \mathcal{X}'_K$, with each sub-stream stored within a brick $M$. Subsequently, its embedding vector $v_i \in \mathbb{R}^{d_1}$

---

**Algorithm 1** Operations

1: **Operation** $Store(x_i)$:
2:    $v_i \leftarrow \mathcal{E}(x_i)$ ; $a_i \leftarrow \mathcal{A}(x_i)$
3:    $M_{\mathcal{H}(x_i)} = M_{\mathcal{H}(x_i)} + v_i \circ a_i$
4: **Operation** $Query(x_i)$:
5:    $v_i \leftarrow \mathcal{E}(x_i)$ ; $a_i \leftarrow \mathcal{A}(x_i)$; $m_i = M_{\mathcal{H}(x_i)}^T a_i$ ;
6:    $s_{\mathcal{H}(x_i)}, s_{\mathcal{H}(x_i)}^{(n)}, s_{\mathcal{H}(x_i)}^{(\alpha)} \leftarrow \mathcal{S}(M_{\mathcal{H}(x_i)})$
7:    $\hat{f}_i' \leftarrow \mathcal{D}(m_i, v_i, s_{\mathcal{H}(x_i)}, s_{\mathcal{H}(x_i)}^{(n)}, s_{\mathcal{H}(x_i)}^{(\alpha)})$

---

**Algorithm 2** Ensemble Decoding

1: **Module** $\mathcal{D}(m_i, v_i, s_{\mathcal{H}(x_i)}, s_{\mathcal{H}(x_i)}^{(n)}, s_{\mathcal{H}(x_i)}^{(\alpha)})$:
2:    $\hat{f}_i' = g_{dec}(m_i, v_i, s_{\mathcal{H}(x_i)})$; $\hat{f}_i'' = min(\frac{m_i}{v_i})$
3:    **if** $s_{\mathcal{H}(x_i)}^{(\alpha)} \notin I_\alpha$ or $s_{\mathcal{H}(x_i)}^{(n)} \leq \beta$: $\hat{f}_i = \hat{f}_i''$
4:    **else**: $\hat{f}_i = \hat{f}_i'$
5:    **return** $\hat{f}_i$

---

is written into $M_{\mathcal{H}(x_i)}$, according to the address vector $a_i \in \mathbb{R}^{d_1 \times d_2}$ as follows: $M_{\mathcal{H}(x_i)} = M_{\mathcal{H}(x_i)} + v_i \circ a_i$, where $\circ$ represents the element-wise multiplication. Similarly, for querying $x_i$, the hash function $\mathcal{H}$ first identifies the relevant memory brick $M_{\mathcal{H}(x_i)}$, which preserves $x_i$'s frequency information. Then, the relevant information $m_i \in \mathbb{R}^{d_1}$ is extracted from $M_{\mathcal{H}(x_i)}$, using $x_i$'s address $a_i$: $m_i = M_{\mathcal{H}(x_i)}^T a_i$[3]. Additionally, we supplement the length of sub-stream into $m_i$, using a counting bucket.

***Memory Scanning*** $(\mathcal{S})$. For any given data stream $\mathcal{X}$, previous sketches typically estimate frequency $f_i$ by data items relevant information $m_i$ retrieved from storage, while failing to consider global characteristics $s$ of the data stream

---

[3]For batch matrix multiplication, vectors need dimensional broadcast and $T$ applies to last two dimensions.

(such as distinct item number $n$ and skewness parameter $\alpha$[4]) as prior knowledge to aid frequency estimation. This oversight arises primarily because handcrafted sketches struggle to infer global stream properties from their compressed storage. Although some works have attempted to infer error distributions based on handcrafted rules to assist in decoding (Ting, 2018; Chen et al., 2021; Liu et al., 2024), such designs are limited and highly dependent on specific rules. In reality, the error of sketches is fundamentally determined by the characteristics of the stream itself, and further using rule-based methods to infer these characteristics from compressed storage is even more challenging. Moreover, designing estimation strategies that consider global characteristics complicates the matter further. Here, we point that, these challenges can be resolved within the MANN architecture of neural sketches through end-to-end training, which could effectively reconstructs global stream characteristics. It not only facilities the reconstruction of both explicit and implicit global stream characteristics $s$ by training scanning network $g_{scan}$ under the supervision of the loss function, but also simultaneously trains a decoding network $g_{dec}$ utilizing global characteristics $s$ for more accurate frequency estimation[2].

Considering the disorderliness of information caused by hash addressing in memory bricks, these stacked pieces of information can essentially be viewed as a set that satisfies permutation invariance (Kimura et al., 2024). Therefore, we employ a simple set prediction network, Deepsets (Zaheer et al., 2017), to implement $g_{scan}$ for scanning and reconstructing global stream characteristics from a specific $M_{\mathcal{H}(x_i)}$. For practical efficiency, we take a subset $M'_{\mathcal{H}(x_i)}$ comprising one-tenth the size of the memory $M_{\mathcal{H}(x_i)}$, as we find this fraction sufficient. This subset $M'_{\mathcal{H}(x_i)}$ is then fed into $g_{scan}$, producing a reconstruction vector $s_{\mathcal{H}(x_i)}$. Among the elements of $s_{\mathcal{H}(x_i)}$, the first two, $s_{\mathcal{H}(x_i)}^{(n)}$ and $s_{\mathcal{H}(x_i)}^{(\alpha)}$, regress to the explicit global characteristics $n$ and $\alpha$ under the supervision of an auxiliary reconstruction loss $\mathcal{L}''$. The remaining elements reconstruct implicit characteristics guided by the main loss $\mathcal{L}'$ described in Section 3.2.

$$s_{\mathcal{H}(x_i)}, s_{\mathcal{H}(x_i)}^{(n)}, s_{\mathcal{H}(x_i)}^{(\alpha)} = \mathcal{S}(M_{\mathcal{H}(x_i)}) = g_{scan}(M'_{\mathcal{H}(x_i)})$$

***Ensemble Decoding***$(\mathcal{D})$. Given a query item $x_i$ and its corresponding $m_i$ and $v_i$, we concatenate $m_i$ and $v_i$ as the foundation, supplementing them with stream characteristics $s_{\mathcal{H}(x_i)}$. These vectors are then fed into a decoding network $g_{dec}$ to obtain a neural prediction $\hat{f}'_i$. Benefiting from the newly designed and more interpretable MANN architecture

of the Lego sketch, we enable the straightforward computation of a rule-based estimate $\hat{f}''_i = \min(m_i/v_i)$. These two estimates are aggregated to produce a final prediction $\hat{f}_i$ based on the estimated skewness $s_{\mathcal{H}(x_i)}^{(\alpha)}$ and item number $s_{\mathcal{H}(x_i)}^{(n)}$, i.e., $\hat{f}_i = \mathcal{D}(m_i, v_i, s_{\mathcal{H}(x_i)}, s_{\mathcal{H}(x_i)}^{(n)}, s_{\mathcal{H}(x_i)}^{(\alpha)})$, as detailed in Algorithm 2.

## 3.2. Training

Since the memory blocks in the memory module $\mathcal{M}$ are independent and are extended by multiple copies of a single memory block, we only need to train the Lego sketch on one memory brick $M$. Overall, we adopted the same self-supervised meta-learning training strategy as previous research (Cao et al., 2024; 2023; Rae et al., 2019), which iteratively trains on a large number of synthetic meta-tasks to acquire the basic sketching capabilities. The meta-tasks are automatically generated based on data streams with different skewness under Zipf distributions, and each meta-task corresponds to storing and querying all items from a synthetic data stream into the memory block $M$. As shown in Algorithm 3 in the appendix, the ultimate goal of the meta-learning process is to optimize the parameters through gradient descent based on the query error of all items.

During the training, we introduce a novel self-guided weighting loss that dynamically assigns weights to different error metrics[5] across different meta-tasks. Previous loss functions $\mathcal{L}_o$ (Cao et al., 2023; 2024; Rae et al., 2019) simply employ an average across a batch $b$ of meta-tasks $\mathcal{T}$ using only learnable weights (LW) (Kendall et al., 2018) for different error metrics. Such methods neglect the variability in task difficulty, causing larger errors to overshadow smaller ones across different meta-tasks. Our method resolves this problem by using a "guide error" from a sketch competitor to weight different error metrics across meta-tasks dynamically, i.e. optimizing $(error)^2/(guide\,error)^2$. Additionally, instead of using a handcrafted sketch as a competitor, the Lego sketch directly use the rule-based $\hat{f}'$ to guide the neural prediction $\hat{f}''$, which is produced by $\mathcal{D}$ simultaneously for training efficiency, leading to the *self-guided weighting loss* $\mathcal{L}'$. This self-guided weighting loss $\mathcal{L}'$ significantly enhances accuracy for large space budgets, effectively addressing the issues of advantage diminished noted in earlier neural sketches[2].

$$\mathcal{L}_o = \frac{1}{|b|} \sum_{\mathcal{T} \in [b]} \text{LW}\left(AAE(\hat{f}'), ARE(\hat{f}'), MSE(\hat{f}')\right)$$
$$\mathcal{L}' = \frac{1}{|b|} \sum_{\mathcal{T} \in [b]} \text{LW}\left(\frac{AAE(\hat{f}')^2}{AAE(\hat{f}'')^2}, \frac{ARE(\hat{f}')^2}{ARE(\hat{f}'')^2}, \frac{MSE(\hat{f}')^2}{MSE(\hat{f}'')^2}\right)$$

Additionally, we incorporate an auxiliary reconstruction loss

---

[4]In real world streams, the frequency distributions exhibit skewed patterns, which are often approximated by the $Zipf$ distributions (Babcock et al., 2002; Yang & Zhu, 2016; Breslau et al., 1999; Adamic, 2000) using a varied parameter $\alpha$ known as skewness as detailed in Appendix D.

[5]AAE$=\frac{1}{n}\sum^n |\hat{f}_i - f_i|$; ARE $=\frac{1}{n}\sum^n |\hat{f}_i - f_i|/f_i$; MSE $=\frac{1}{n}\sum^n |\hat{f}_i - f_i|^2$

$\mathcal{L}''$ in $\mathcal{S}$, constituting the final loss $\mathcal{L}$:

$$\mathcal{L} = \mathcal{L}' + 0.1 \times \mathcal{L}'', \text{ where}$$

$$\mathcal{L}'' = \frac{1}{|b|} \sum_{\mathcal{T} \in [b]} \text{LW}\left( MSE(s_{\mathcal{H}(x_i)}^{(n)}, n), MSE(s_{\mathcal{H}(x_i)}^{(\alpha)}, \alpha) \right)$$

### 3.3. Derivative with Lego Sketch as Core Structure

As a representative of derivatives (Zhou et al., 2018; Roy et al., 2016; Hsu et al., 2019; Aamand et al., 2024; Zhao et al., 2023; Gao et al., 2024b), the elastic sketch derivative (Yang et al., 2018) comprises two primary components: a heavy part with filtering buckets for high-frequency items and a light part, typically a CM-sketch, for low-frequency items. This design reduces estimation errors by effectively segregating high- and low-frequency items during data storage. We present a case study of applying the framework of elastic derivative with the Lego sketch as the core. Results from Sections 5.2 and 5.3 demonstrate its superior accuracy and robustness, demonstrating the potentials of the Lego sketch in future advancements.

## 4. Analysis

### 4.1. Domain Scalability Analysis

As detailed in the proof provided in Appendix E, Theorem 4.1 demonstrates that embedding vectors from different data domains adhere to the same distribution, confirming that the normalized multi-hash embedding technique is domain-agnostic. In Section 5.2, the Lego sketch deploys across datasets from five distinct domains, as shown in Table 2, achieves consistently high accuracy.

**Theorem 4.1.** *Across all data items $\{x_i\}$ within any data domain $\mathbb{X}$, the embedding vectors $\{v_i\}$ generated by the normalized multi-hash embedding technique exhibit the same distribution.*

### 4.2. Memory Scalability Analysis

Here, the impact of memory scalability on generalizability is assessed by quantifying the gap between the sub-skewness $\alpha'$ of sub-stream $\mathcal{X}'$ in each brick and the overall skewness $\alpha$ of the entire stream $\mathcal{X}$. Assuming the frequency ranking of an item $x_i$ in $\mathcal{X}$ is denoted by $r_i$, and its corresponding sub-ranking in $\mathcal{X}'$ is denoted by $r_i'$, we can derive Theorem 4.2, the proof of which is in Appendix F.

**Theorem 4.2.** *Given $K$ memory bricks, the sub-skewness $\alpha'_{r_i'}$ around $r_i$ in sub-stream $\mathcal{X}'$ is:*

$$\alpha'_{r_i', K}(D, r_i) = \alpha \log(1 + \tfrac{D}{r_i}) / \log(1 + \tfrac{1}{r_i'}), \text{ where}$$

$$D \sim G(1/K), \ (r_i - r_i') \sim NB(r_i', 1/K)$$

*$D$ denotes the distance of ranking $r$ between two data items, which are adjacent on the sub-ranking $r'$ in same $\mathcal{X}'$. Con-*

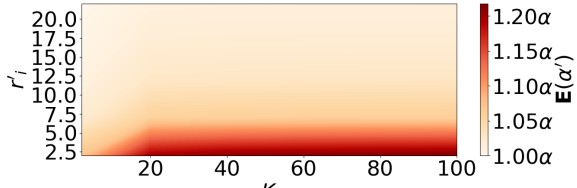

Figure 5: Sub-skewness $\alpha'$

Table 2: Real Datasets Summary

| Name | Aol | Lkml | Kosarak | Wiki | Webdocs |
|---|---|---|---|---|---|
| $n$ | 197,790 | 242,976 | 41,270 | 9,379,561 | 5,267,656 |
| $N$ | 361,115 | 1,096,439 | 8,019,015 | 24,981,163 | 299,887,139 |
| Domain | Word | Comm. | Click | Edit | Doc. |

*sequently, the expected sub-skewness is as follows:*

$$\boldsymbol{E}(\alpha'_{r_i', K}) = \sum^D \sum^{r_i} \alpha'_{r_i', K}(D, r_i) \boldsymbol{P}(D|K) \boldsymbol{P}(r_i | r_i', K)$$

As numerical simulation in Figure 5, upon excluding the top ten high-frequency items, the sub-skewness $\alpha'$ surrounding other items closely approximates the overall skewness $\alpha$. Given the massive volume of items in data streams, the Lego sketch thereby effectively transfers learned skewness patterns to each sub-stream, thus improving estimation accuracy. Also the experimental evidence in Section 5.2, including memory resizing exceeding 1400-fold: from single 100KB brick to a maximum of 140MB in Figure 6(c) (Wiki Dataset), underscores its powerful memory scalability.

### 4.3. Error Analysis

Here, we give the error bound for the rule-based estimation $\hat{f}_i''$ in Lego sketch as Theorem 4.3, and the detailed proof is provided in Appendix G.

**Theorem 4.3.** *The error of $\hat{f}_i''$ for item $x_i$ is bounded by:*
$$\boldsymbol{P}(|\hat{f}_i'' - f_i| \geq \epsilon \times N) \leq (\epsilon \times d_2)^{-1}$$

In general, formulating an error bound for a pure neural architecture poses challenges, leading to a scarcity of error analysis in prior research. The above analysis shows a wider margin compared to handcrafted sketch boundaries. This discrepancy primarily stems from the normalization within the embedding module, causing non-independent values in $v_i$. Future work may explore adjustments to the normalization approach or modeling the distribution of $v_i$ to further strengthen the error bound.

## 5. Experiment

### 5.1. Setup

**Datasets.** Firstly, we employ five real datasets indicating irregular distributions in real applications, Aol (Pass et al.,

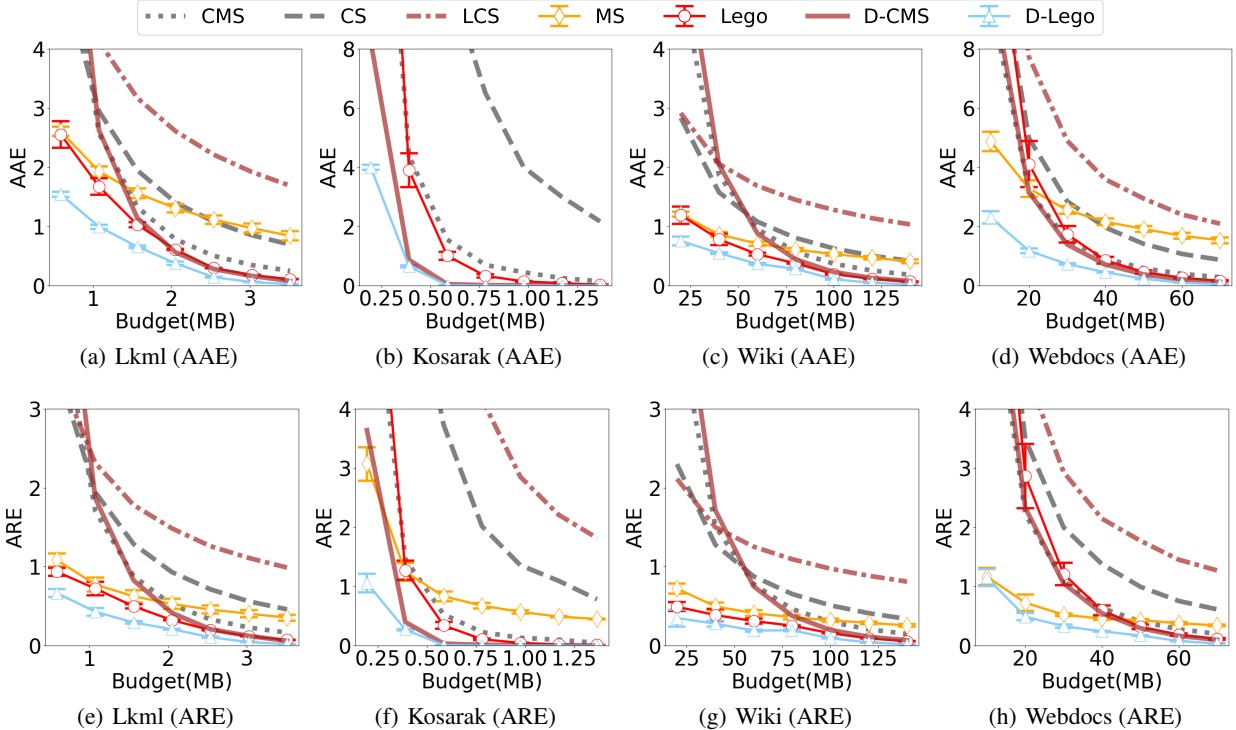

Figure 6: Errors on Four Real Datasets.

2006), Lkml (Homscheid et al., 2015), Kosarak (Bodon, 2003), Wiki (Kunegis, 2013), and Webdocs (Lucchese et al., 2004). These five datasets are widely used in sketch literatures (Roy et al., 2016; Cao et al., 2023; Hsu et al., 2019; Aamand et al., 2024) and encompass a variety of data stream domains and scales for comprehensive evaluation, as shown in Table 2. [6] Secondly, we generate six synthetic data streams, each comprising $100,000$ distinct data items, conforming the $Zipf$ distribution within the skewness $\alpha$ range of $[0.5, 1.5]$ to evaluate the robustness across different skewnesses comprehensively. All datasets are evaluated using two most widely used error indicators (Cao et al., 2023; Yang et al., 2018; Zhou et al., 2018; Roy et al., 2016; Deng & Rafiei, 2007), specifically Average Absolute Error (AAE) and Average Relative Error (ARE)[5].

**Baselines.** In our experiments, we select a broad range of baselines for comprehensive evaluations. Key among these are the CM-sketch (CMS) (Cormode & Muthukrishnan, 2005) and C-sketch (CS) (Charikar et al., 2002), the most widely used sketches, foundational to subsequent derivatives (Roy et al., 2016; Yang et al., 2018; Zhou et al., 2018; Hsu et al., 2019). We also included the learned augmented C-

sketch(LCS) (Aamand et al., 2024) which only utilizes prior knowledge of $n$ without orthogonal add-ons, making it a close variant to the core sketches. Additionally, we consider the latest neural sketch, the meta-sketch (MS) (Cao et al., 2023; 2024), as a significant baseline, which is renowned for exceptional accuracy in small space budgets. Also, we incorporate scalable memory module into the meta-sketch for memory scalability, facilitating a comprehensive comparison in different budgets. Finally, we evaluate the elastic derivative with Lego sketch, D-Lego, as detailed in Section 3.3. Accordingly, the elastic derivative with CMS, i.e., D-CMS (Yang et al., 2018), is considered the competitor.

**Parameters.** The Lego sketch is pretrained with $4$ million meta-tasks within the regular skewness range of $[0.5, 1.0]$. It uses $8$-layer Deep Sets networks for the $g_{scan}$ and $g_{dec}$, featuring a maximum hidden layer size of $32$ and employing LeakyReLU for layer connectivity. The initial learning rate is set at $0.001$, decreasing linearly to $0.0001$. The aggregation threshold $\beta$ is consistently set at $10,000$. For all CMS and CS, we employ three hash functions, as standard practice suggests(Aamand et al., 2024; Yang et al., 2018). Further details are available in the appendix C and code provided in supporting materials.

**Budget.** In line with previous works (Cao et al., 2023; 2024; Rae et al., 2019), the space budget $B$ is determined by the total size of $K$ memory bricks $M$, which in our experiments is 100KB per brick $M(d_1 = 5, d_2 = 5120)$. The budget

---

[6]Following previous works, we also tested the Aol dataset, where Lego Sketch demonstrated a significant advantage. During the review process, the reviewers pointed out its anonymization flaw, (see Aol log release ); therefore, we have excluded the results from experiments and ablation studies in our revised manuscript.

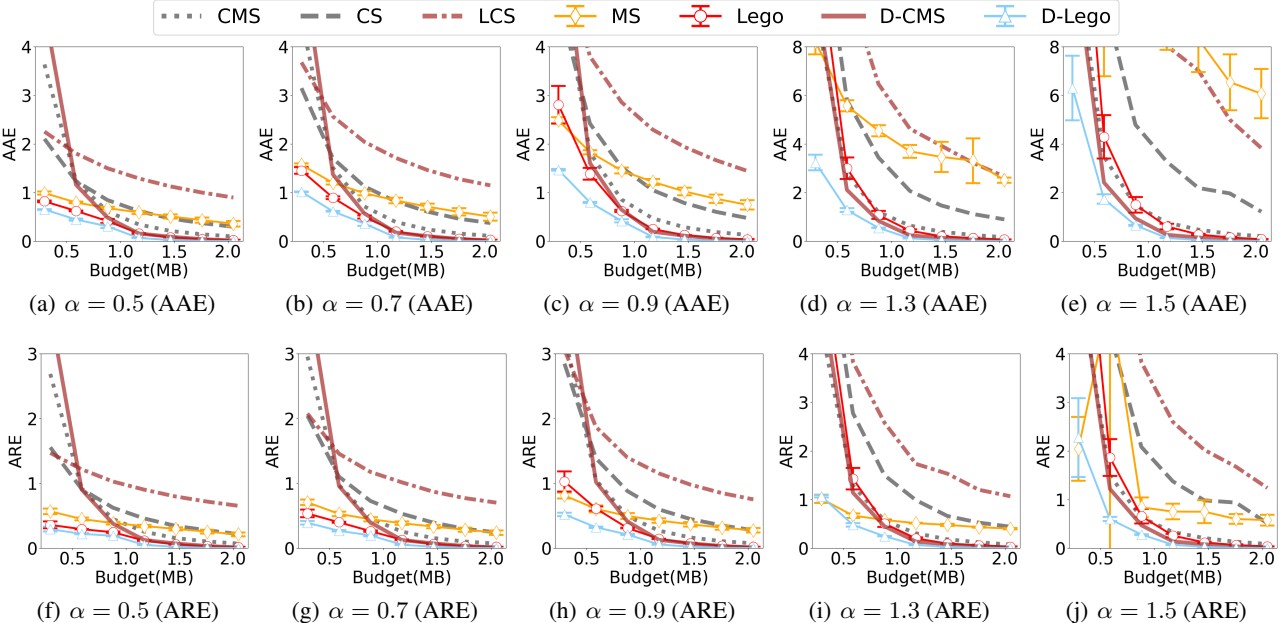

Figure 7: Synthetic Datasets

allocated for filtering buckets in D-CMS and D-Lego is set to one-fourth of $B$ (Yang et al., 2018).

## 5.2. Accuracy

In this section, we conduct comprehensive evaluations using different space budgets on five real datasets in Figure 6, where the Lego sketch consistently outperforms all competitors in terms of AAE and ARE, offering superior space-accuracy trade-off. Taking the commonly used Lkml dataset in Figures 6(a) and 6(e) as an example, we analyze the space-accuracy trade-off characteristics of the Lego sketch and its competitors. CMS and CS, recognized as the most popular handcrafted sketches, demonstrate CMS's superior accuracy in large space budgets, while CS performs better in smaller space budgets. LCS, an adaptation of CS with prior knowledge of $n$, reduces errors in smaller budgets at the expense of accuracy in larger budgets. MS, leveraging a purely neural architecture, significantly outperforms the aforementioned sketches in smaller spaces. However, at larger budgets, such as at 3.2MB, MS falls behind handcrafted sketches.

Notably, our Lego sketch, as a novel neural sketch, further addresses the drawbacks of meta sketches in large space budgets and comprehensively reduces estimation errors. Specifically in Figure 6(e), with the smallest budget of 0.6 MB, the ARE of the Lego sketch is only 0.93, which represents just 85% of the error of the neural architecture MS, and 21% and 26% of the errors of the traditional CMS and CS, respectively. At the largest budget of 3.6 MB, the ARE is 0.074, further reducing to just 21%, 46%, and 16% of the errors of MS, CMS, and CS, respectively. The similar

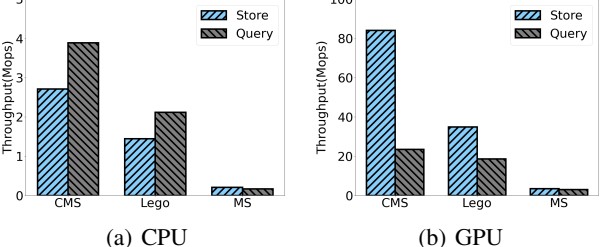

Figure 8: Throughput

advantage is observed in terms of AAE. For example, as illustrated in Figure 6(a), at budgets of 0.6 MB and 3.6 MB, the AAE of the Lego sketch is only 97% and 13% of that of MS. These results consistently demonstrate a significant reduction in estimation errors across all budget levels, underscoring the effectiveness of our approach.

Moreover, the Lego sketch within framework of elastic derivative, abbreviated as D-Lego, is a case study in the derivative category, exhibiting a better space-accuracy trade-off. For example, at the 3.6MB budget, its AAE dramatically reduces to an tiny value of 0.03, demonstrating significantly enhanced accuracy. Additionally, this version notably outperforms the elastic derivative with the traditional CM-sketch, abbreviated as D-CMS, maintaining a consistent lead of approximately 5-folds to 3-folds. These results highlights the immense potential of integrating derivative strategies with the Lego sketch as core sketch in the future.

## 5.3. Robustness under Distributional Shift

To verify the robustness of the Lego sketch against potential distributional shifts across varying degrees of skewness, we

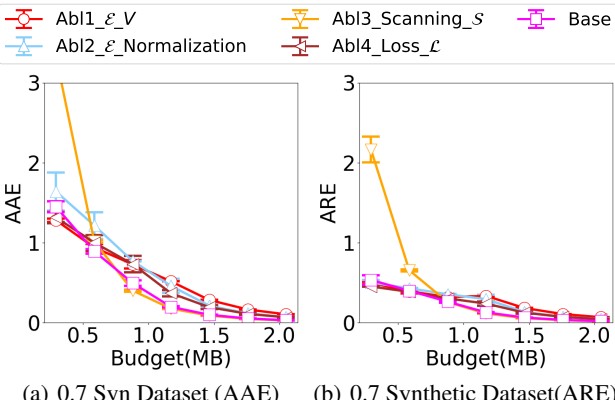

(a) 0.7 Syn Dataset (AAE)   (b) 0.7 Synthetic Dataset(ARE)

Figure 9: Ablation Study

also assess its accuracy under synthetic streams conforming to Zipf distributions with different skewness $\alpha$. Due to space constraints, some experimental results are shown in Appendix A. As shown in Figure 7, as the skewness $\alpha$ increases from a relatively low value of 0.5 to a high value of 1.5, the estimation errors of all sketches progressively rise. The meta-sketch and LCS exhibit poorer robustness; for instance in Figure 7(e), at an extreme skewness of 1.5, their absolute errors predominantly exceed 8, surpassing the upper bound depicted even in large budgets. In contrast, the Lego sketch demonstrates strong estimation robustness with the skewness $\alpha$ increasing, ultimately performing comparably to the CM-sketch and surpassing all other baselines. Notably, the elastic derivative with Lego sketch (i.e. D-Lego) shows the best robustness, underscoring the immense potential of future research in derivatives based on Lego sketch.

### 5.4. Throughput

We evaluate the throughput of Lego Sketch's *Store* and *Query* operations on the large-scale Wiki dataset using both CPU and GPU platforms. CMS, with its simple structure and high throughput, and MS serve as baselines for comparison. As shown in Figure 8, the overall throughput of the Lego sketch is of the same magnitude as that of CMS, reaching millions on CPU and tens of millions with GPU acceleration. In contrast, compared to MS, the Lego sketch, with its more effective MANN architecture, significantly surpasses MS. For instance, the throughput for storing is up to 6.90 times higher and for querying is 12.47 times higher, on CPU. This shows the Lego sketch's capability to deliver accurate estimation while efficiently processing streams.

### 5.5. Ablation Study

Figure 9 presents ablation studies conducted on two datasets, evaluating the impact of four techniques across three key modules of the Lego Sketch framework. The results are summarized as follows:

**Embedding Module** $\mathcal{E}$**.** In the first and second ablation tests, we remove the learnable vector $\mathcal{V}$ and normalization operation in the normalized multi-hash embedding, respectively. When using a fixed $\mathcal{V}$ for embedding instead of a learnable vector, the error increases significantly under higher budgets, highlighting the necessity of end-to-end training for accuracy prediction scenarios. Moreover, removing the normalization operation consistently leads to higher errors across all budgets, underscoring the role of $L_1$ stability in maintaining stable memory increments for embedding.

**Scanning Module** $\mathcal{S}$**.** In the third ablation test, we removed the scanning module, which was newly introduced by the Lego Sketch. This change directly causes a significant rise in errors under small budgets, indicating that the novel scanning module $\mathcal{S}$ effectively captures the global characteristics of data streams through end-to-end training. Consequently, it significantly improves frequency estimation accuracy for neural sketching in low-budget scenarios.

**Self-guided Weighting Loss** $\mathcal{L}$**.** The final ablation experiment replaces the self-guided weighting loss $\mathcal{L}'$ with the conventional loss $\mathcal{L}_o$ used in previous work. We observe that without the self-guided weighting optimization, the accuracy of Lego sketch deteriorates under larger budgets, lead to higher errors in such scenarios. This demonstrates that the self-guided weighting loss effectively guides the model by dynamically reweighting different meta-tasks according to varying task difficulty during training, enabling a better space-accuracy trade-off.

## 6. Conclusion

In this work, we propose the Lego sketch, a novel neural sketch crafted to overcome the scalability and accuracy challenges encountered by existing neural sketches in real-world stream applications. Mirroring the sturdy and modular nature of Lego bricks, the Lego sketch pioneers a scalable memory-augmented neural network capable of adapting to various data domains, space budgets, and offers favorable space-accuracy trade-off, with ease and efficacy. Within the proposed framework of the Lego sketch, a suite of advanced techniques, including hash embedding, scalable memory, memory scanning, and a tailored loss function, collectively ensure the estimation accuracy of the Lego sketch. Extensive experiments demonstrate that the Lego sketch outperforms existing handcrafted and neural sketches, while its potential integration with orthogonal add-ons holds promise in facilitating data stream processing.

## Acknowledgements

We thank the reviewers for their constructive feedback during the review process. This work was supported in part by the National Natural Science Foundation of China under Grant 62472400, Grant 62072428, Grant 62271465, and in part by the Suzhou Basic Research Program under Grant SYG202338.

## Impact Statement

This paper presents work whose goal is to advance the field of Machine Learning. There are many potential societal consequences of our work, none which we feel must be specifically highlighted here.

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

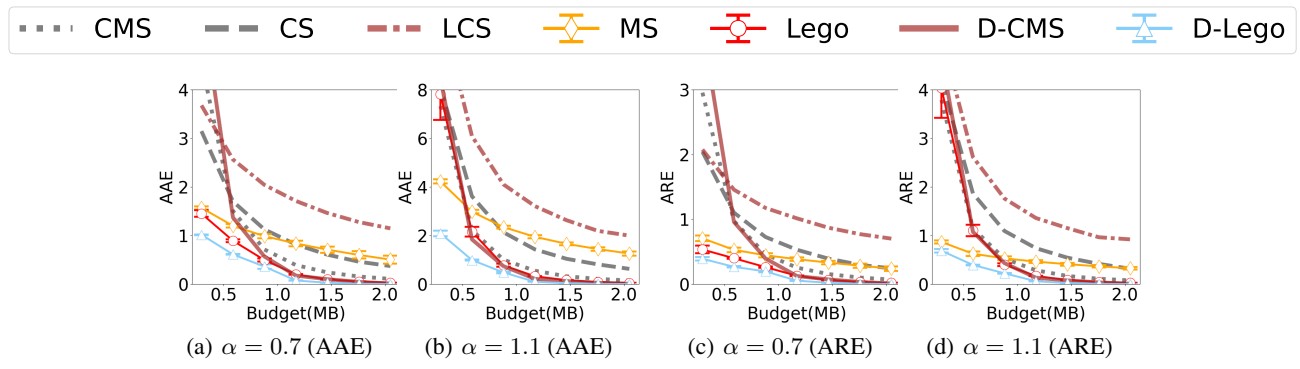

Figure 10: Absent Results of Synthetic Datasets

---

**Algorithm 3** Training Algorithm

---

1: **Input:** Model($K = 1$) with all learnable parameters $\theta$
2: **while** not reaching the max step **do**
3:     Generating a batch $b$ of meta-tasks
4:     **for** meta task $\mathcal{T} \in b$ **do**
5:         Clear the memory $M$
6:         *Store* all items in $\mathcal{T}.support\ set$
7:         *Query* all frequencies of distinct items in $\mathcal{T}.query\ set$
8:     **end for**
9:     Calculate $\mathcal{L}$ across all frequencies estimations of $b$
10:     Backward propagate through $\partial \mathcal{L} / \partial \theta$
11:     Update all parameter $\theta$
12: **end while**

---

## A. Additional Experimental Results

Figures 10 are provided herein, excluded from the main text due to space constraints.

## B. Limitations

Lego sketch, like other existing neural sketches (Cao et al., 2024; 2023), faces challenges in error analysis due to the limited interpretability of deep learning techniques (MANN) they employ. While we provide an initial error analysis for Lego sketch, it does not yet match the tightness of the error bounds achieved by handcrafted sketches. Future work should focus on improving error analysis techniques for neural sketches to address this gap. Furthermore, given the broad applicability of sketching techniques, extending the scalable and accurate architectural innovations of Lego sketch to related areas such as graph stream summarization (Tang et al., 2016; Feng et al., 2024; Zhao et al., 2025) or vector compression (Ivkin et al., 2019; Spring et al., 2019; Gui et al., 2023; Zhang et al., 2023) represents a promising direction for future research.

## C. Parameter settings

In the embedding module $\mathcal{E}$, the dimension of the learnable vector $V$ is set to 80 and all values in $V$ are constrained to $[\epsilon, 1]$ to ensure numerical stability during decoding processes, where $\epsilon = 0.001$. $I_n$ is set within $[1000, 50000]$ to encompass a broad spectrum of data stream scenarios. $I_N$, on the other hand, varies up to 10 times the minimum stream length permissible by current n and alpha, ranging from $[minlength, 10 \times minlength]$. All experiments run at a NVIDIA DGX workstation with CPU Xeon-8358 (2.60GHz, 32 cores), and 4 NVIDIA A100 GPUs (6912 CUDA cores and 80GB GPU memory on each GPU). The training time for a single Lego sketch is approximately 48 hours, and the relevant code is provided in the supporting materials.

## D. Zipf Distribution

**Definition D.1.** The $Zipf(\alpha)$ distribution describes the pattern between the frequency ratio $p_i$ and the ranking $r_i$ (counting from 1 to n) of items as below:

$$p_i = \frac{C}{r_i^\alpha}, \ where \ C = (\sum_{i=1}^{n} r_i^{-\alpha})^{-1}.$$

The $\alpha$ is a parameter indicating the skewness of $Zipf$ distribution among the data stream. Thus given the ranking $r_i$ of the item $x_i$ and the stream length $N$, its frequency can be calculated as $f_i = Np_i = \frac{NC}{r_i^\alpha}$.

## E. Proof of Theorem 4.1

**Theorem E.1.** *Across all data items $\{x_i\}$ within any data domain $\mathbb{X}$, the embedding vectors $\{v_i\}$ generated by the normalized multi-hash embedding technique exhibit the same distribution.*

*Proof.* We can divide the normalized multi-hash embedding technique $\mathcal{E}(x_i)$ into two stages as follows.

$$v_i' = (V_{\mathcal{H}_1(x_i)}, V_{\mathcal{H}_2(x_i)}, \ldots, V_{\mathcal{H}_{d_1}(x_i)})$$

$$and$$

$$v_i = v_i' / \sum_{j=1}^{d_1} v_{ij}'$$

Given that the hash functions $\mathcal{H}$ uniformly distribute outputs for any given input, each component of the vector $v_i'$ associated with any input $x_i$ is independently and identically distributed (i.i.d.), with each $P[v_{ij}' = V_k] = \frac{1}{d_1}$. Consequently, for any data item, the vector $v_i$ can be derived by normalizing $v_i'$ as $v_i = v_i'/\sum_{j=1}^{d_1} v_{ij}'$, under the same probability condition $P[v_{ij}' = V_k] = \frac{1}{d_1}$. Therefore, across all data items $x_i$ within any data domain $\mathbb{X}$, the resulting embedding vectors $v_i$ adhere to a consistent distribution, as determined by the normalized multi-hash embedding technique. $\square$

## F. Proof of Theorem 4.2

**Theorem F.1.** *Given $K$ memory bricks, the sub-skewness $\alpha_{r_i'}'$ around $r_i$ in sub-stream $\mathcal{X}'$ is as follows:*

$$\alpha_{r_i', K}'(D, r_i) = \alpha \log(1 + \frac{D}{r_i}) / \log(1 + \frac{1}{r_i'})$$

$$where \ D \sim G(1/K), (r_i - r_i') \sim NB(r_i', 1/K)$$

*$D$ denotes the distance of ranking $r$ between two data items, which are adjacent on the sub-ranking $r'$ in same $\mathcal{X}'$. Consequently, the expected sub-skewness is as follows:*

$$\boldsymbol{E}(\alpha_{r_i', K}') = \sum_{D}^{D} \sum_{}^{r_i} \alpha_{r_i', K}'(D, r_i) \boldsymbol{P}(D|K) \boldsymbol{P}(r_i | r_i', K)$$

$$= \alpha \sum_{D=1}^{\infty} \sum_{r_i=r_i'}^{\infty} \frac{\log(1 + \frac{D}{r_i})}{\log(1 + \frac{1}{r_i'})} \frac{(K-1)^{r_i + D - r_i' - 1}}{K^{(D+r_i)}} \binom{r_i - 1}{r_i' - 1}$$

*Proof.* When the Zipf distribution $p_i = \frac{C}{r_i^\alpha}$ is transformed into a log-log scale, the relation between $\log p_i$ and $\log r_i$ becomes linear, i.e. $\log p_i = \log C - \alpha \log r_i$. Here, $\alpha$ manifests as the negative slope of the line, illustrating the skewness in of data stream frequencies. We therefore derive the $\alpha_{r', K}'$ by inferring the negative slope between frequencies $\log p_i$ and the sub-ranking $\log r_i'$. Consider a random variable $D$ that denotes the distance of rank $r$, between two data items loaded into the same sketch which are adjacent according $r'$. Without losing generality, we define two items $x_i$ and $x_j$ satisfying the aforementioned adjacency relationship. Then, we can get the following correspondence:

$$r_j = r_i + D \ and \ r_j' = r_i' + 1$$

In that case, the negative slope on a log-log scale can be calculated in the following way to characterize the $\alpha'_{r'_i, K}$:

$$\alpha'_{r'_i} = -\frac{\log(p_j) - \log(p_i)}{\log(r'_j) - \log(r'_i)} = -\frac{\log(\frac{C}{(r_i + D)^\alpha}) - \log(\frac{C}{(r_i)^\alpha})}{\log(r'_i + 1) - \log(r'_i)} = \alpha \frac{\log(1 + \frac{D}{r_i})}{\log(1 + \frac{1}{r'_i})}$$

Given the vast number of data items in the data stream, from the perspective of a specific sub-stream, the procedure of uniformly distributing data items across $K$ matrices using a hash function can be approximated as a Bernoulli process with a probability of $\frac{1}{K}$. Thus $D$ obeys a geometric distribution with parameter $\frac{1}{K}$, i.e. $D \sim G(\frac{1}{K})$, and $r_i - r'_i$ obeys a pascal distribution, i.e. $(r_i - r'_i) \sim NB(r'_i, \frac{1}{K})$. $\qquad\square$

## G. Proof of Theorem 4.3

**Theorem G.1.** *The error of rule-based estimation $\hat{f}_i''$ for item $x_i$ is bounded by:* $\boldsymbol{P}(|\hat{f}_i'' - f_i| \geq \epsilon \times N) \leq (\epsilon \times d_2)^{-1}$

*Proof.* Let $v_i^k$ be the k-th value of the embedding vector for the item $x_i \in \mathbb{R}^{d_1}$ and $\mathcal{I}_{i,j,k}$ be a Bernoulli random variable indicating if the item $x_i$ and the item $x_j$ is addressed to the same address across $d_2$ slots for the $v_i^k$, then:

$$\mathbf{E}[\mathcal{I}_{i,j,k}] = d_2^{-1}$$

Recalling the storing operation shows that the error of $\hat{f}_i''$ all comes from the address conflict between different $v_i^k$, thus:

$$\hat{f}_i'' \geq f_i \text{ and } \forall k \ (\hat{f}_i'' - f_i \leq \sum_{j=1, j\neq i}^{n} \frac{\mathcal{I}_{i,j,k} * f_j * v_j^k}{v_i^k})$$

$$\hat{f}_i'' - f_i \leq \frac{1}{d_1} \sum_{k=1}^{d_1} \sum_{j=1, j\neq i}^{n} \frac{\mathcal{I}_{i,j,k} * f_j * v_j^k}{v_i^k}$$

Considering the normalization operation acting on the embedding vectors, the expectation of the error can be expressed as:

$$\mathbf{E}[\hat{f}_i'' - f_i] \leq \frac{1}{d_1} \sum_{k=1}^{d_1} \sum_{j=0, j\neq i}^{n} \mathbf{E}[\mathcal{I}_{i,j,k}] * f_j * \mathbf{E}[\frac{v_j^k}{v_i^k}] \leq \frac{N}{d_2}$$

According to Markov's inequality we have:

$$\mathbf{P}(\hat{f}_i'' - f_i \geq \epsilon * N) \leq \frac{\mathbf{E}[\hat{f}_i'' - f_i]}{\epsilon N} \leq (\epsilon * d_2)^{-1}$$

$\qquad\square$

