# OpenReview forum: "Lego Sketch: A Scalable Memory-augmented Neural Network for Sketching Data Streams"
_ICML.cc/2025/Conference — ICML 2025 poster_

### Official Review · Reviewer_o8J2 · 2025-03-07

**Overall Recommendation:** 3

**Summary:**

The paper presents a new approach to solving the frequency estimation problem on a data stream using small space. The idea is to use a neural network to create a sketch that is tuned to the distribution of the input stream. Experiments show that the average frequency error of the approach is lower than competing randomized sketches, including a recent learning-based approach.

**Claims And Evidence:**

The claim that the proposed sketch outperforms existing sketches can be questioned since there is no comparison to the class of counting-based sketches such as Misra-Gries.

The authors have a point that randomized linear sketches are usually not compared to deterministic sketches in the literature, and this can be justified in some application scenarios where the power of linear sketches is needed. I think the suggestion to mention deterministic sketches such as MG, and include a discussion of when they are applicable, is acceptable.

**Essential References Not Discussed:**

I would like to see a comparison to Misra-Gries and related sketches, e.g., the implementation in the DataSketches library (https://arxiv.org/abs/1705.07001). Such algorithms should be superior for large domains since items with no occurrences will alway have an estimate of zero.

**Experimental Designs Or Analyses:**

The experiments seem sound.

**Methods And Evaluation Criteria:**

The choice of error measures can be debated since an average is taken over the domain from which stream elements come. This error measure becomes questionable when the size of the domain is much larger than the size of the sketch since small errors on non-occurring elements will tend to dominate.

**Other Comments Or Suggestions:**

The comment (page 4) that sketches typically do not consider skew is misleading: As shown by Charikar, Chen, and Farach-Colton CountSketch performs better for skewed distributions. Similar results are known for Count-Min sketch.

The use of three hash functions for CS is further supported by https://icml.cc/virtual/2021/poster/9691

**Other Strengths And Weaknesses:**

Nothing particular comes to mind.

**Questions For Authors:**

Would your approach be able to support merging of sketches? This would make it applicable outside of streaming settings.

Update after rebuttal:
Regarding mergeability I am not sure I managed to convey to the authors what my concern was. If the same models are used to two sketches they are of course mergeable, but in general you may want to use different models for different parts of a dataset, to exploit that they have different characteristics. However, merging sketches having different models seems difficult. It would be good to have some discussion of this aspect.

**Relation To Broader Scientific Literature:**

While the paper makes a detailed comparison to related, hashing-based sketches, it is known that counting-based sketches are often superior for insertion-only streams, especially for datasets that are skewed and where the distribution over time is relatively stable. Furthermore, these sketches are mergeable.

**Theoretical Claims:**

I did not check the proofs in the appendix but have no reason to suspect any errors.

---

> ### Author Rebuttal · Authors · 2025-04-01
>
> Thank you for your feedback! We have addressed your concerns as below and welcome any further discussion should you have additional questions.
>
> ## Comparison to insertion-only streams algorithm like Misra-Gries
>
> ### Reply to
> >...counting-based sketches are often superior for insertion-only streams...
>
>
> Counter-based methods (e.g., Misra-Gries) and sketch-based methods (e.g., Count-Min Sketch, Lego Sketch) represent two distinct lines of research in streaming algorithms, as classified by Cormode and Hadjieleftheriou in their foundational work "Finding Frequent Items in Data Streams" (VLDB 2008).
>
> So, we respectfully argue that directly comparing sketch-based methods (e.g., Count-Min Sketch, Lego Sketch) with counter-based methods (e.g., Misra-Gries Algorithm) is inappropriate, due to their inherently different design goals: counter-based methods identify frequent items in insertion-only streams with minimal overhead, while sketch-based methods focus on probabilistic frequency estimation for dynamic streams (supporting deletions/weights). This methodological separation is well-recognized in the literature (see papers in reference for details), where sketch-based studies explicitly avoid benchmarking against counter-based methods, due to their fundamentally distinct problem formulations and guarantees.
>
> The key differences between the two methods are summarized in the table below, based on Cormode’s paper.
>
>
> |**Aspect**|**Counter-Based Methods**| **Sketch-Based Methods**|
> |-|-|-|
> |**Stream Type**|Insertion-only streams.| Dynamic streams (insertions, deletions, weights). |
> |**Primary Use Case**| Identify frequent items (heavy hitters).| Estimate frequency of any item (supports queries). |
> |**Data Structure**| Explicit counters for tracked items.| Compact 2D array with hash-mapped counters.|
> |**Handles Deletions**|No.| Yes.|
> |**Supports Weights**|Limited.| Yes (positive/negative weights).|
> |**Flexibility**|Optimized for tracking frequent items.|Extendable to quantiles, entropy, inner products.|
> |**Examples**| Misra-Gries, SpaceSaving, LossyCounting.|Count-Min Sketch, CountSketch.|
> |**Strengths** | High precision, low space, fast updates. |Versatile, handles deletions, broader applications.|
> |**Weaknesses**|Limited to insertions. |Higher space/time costs;|
> |**Common Error Metric**|ARE (Average Relative Error), AAE (Average Relative Error)  | ARE, AAE  |
>
>
>
> ## Methods And Evaluation Criteria:
> ### Reply to
> > ... This error measure becomes questionable ... since small errors on non-occurring elements will tend to dominate.
>
> We respectfully argue that AAE(Average Absolute Error) and ARE(Average Relative Error) have been well-established standards since the early 2000s (see papers in reference). To ensure fair evaluation and comparison, we adhere to these established practices.
>
> Moreover, non-occurring elements are often excluded in prior sketch-based literature for two reasons: 1) They lack true frequency, making ARE computation infeasible; 2) Hash collisions introduce similar absolute errors for both occurring and non-occurring elements, leading to minimal impact on AAE.
>
>
> |CM Sketch |Syn stream (skewness$ \alpha=0.6$) |Syn stream (skewness$\alpha=0.8$)|Syn stream (skewness$\alpha=1.0$)|Syn stream (skewness$\alpha=1.2$)|  Syn stream (skewness$\alpha=1.4$)|
> |-|-|-|-|-|-|
> |AAE 10K occurring elements|2.98|1.87|1.33|1.04| 0.89|
> |AAE 10K non-occurring elements|3.09|1.88|1.31|1.05| 0.89|
> |ARE 10K occurring elements|0.70|0.49|0.41|0.40| 0.36|
> |ARE 10K non-occurring elements (Infeasible)|n/a|n/a|n/a|n/a| n/a|
>
>
> ## Other Comments Or Suggestions:
> ### Reply to
> > The comment (page 4) that sketches typically do not consider skew is misleading...
>
> Thank you for pointing this out. We agree that our original phrasing could be clearer in explaining how existing sketches handle skewness.
>
>
>
> While traditional sketches like CM-Sketch and C-Sketch exhibit different performance under skewed data, this behavior results from their static design rather than active skewness detection. In contrast, Lego Sketch introduces a scanning module that dynamically estimates stream skewness and uses it as prior knowledge to enhance accuracy, as shown in our experiments.
>
>
> We will revise the text for clarity:
>
> "Handcrafted sketches face challenges in inferring global stream properties (e.g., distinct item count, skewness) from their compressed storage. While some perform differently under varied skewness, this stems from static design rather than adaptive inference."
>
> ## Questions For Authors:
> ### Reply to
> > Would your approach be able to support merging of sketches?
>
>  Like Meta Sketch, Count-Min Sketch, and other sketch-based methods, our Lego Sketch maintains additive properties that fully support sketch merging operations. We will explicitly clarify this consistency in the revised manuscript.

---

> > ### Comment · Reviewer_o8J2 · 2025-04-01
> >
> > Thanks for your response.
> >
> > Note that counter based sketches do support weights as well as deletions (though accuracy depends on the number of deletions). At least there should be a theoretical discussion of what situations probabilistic sketches give a benefit over deterministic sketches.
> >
> > Considering merging I wonder: Suppose that Alice and Bob independently construct Lego sketches of datasets A and B, respectively. How can these sketches be merged?

---

> > > ### Author Response · Authors · 2025-04-02
> > >
> > > Thank you for your prompt reply. I think your concerns about the relationship between counter- and sketch-based method have been addressed by Berinde and Cormode's work, "Space-optimal heavy hitters with strong error bounds (TODS 2010)" and your concerns about sketch mergability have been addressed by Agarwal and Cormode's work, "Mergeable Summaries (TODS 2013)".
> > >
> > > ## Reply to
> > > >  Note that counter based sketches do support weights as well as deletions (though accuracy depends on the number of deletions). At least there should be a theoretical discussion of what situations probabilistic sketches give a benefit over deterministic sketches.
> > >
> > > **Counter- vs. Sketch-based Method.** As discussed in Cormode and Berinde's works, counter- and sketch-based methods are with different problem scopes. Counter-based method specializes in tasks like heavy hitters, where sketch-based methods are more general in supporting frequency estimation for arbitrary items.
> > >
> > > For reference, Berinde's work highlights their difference, " A key distinction of sketch algorithms (to counter-based methods) is that they allow both positive and negative updates (where negative updates can correspond to deletions, in a transactional setting, or simply arbitrary signal values, in a signal processing environment)...So, although our results show that counter algorithms are strictly preferable to sketches when both are applicable, there are problems that are solved by sketches that cannot be solved using counter algorithms."
> > >
> > > **Performance on deletions and negative weights.** Counter-based methods typically requires two separate instances paired with the triangle inequality (see Anderson's work "A high-performance algorithm for identifying frequent items in data streams, IMC 2017"). Their error scales with $\sum |f_i|$, whereas sketch-based methods achieve error proportional to $\sum f_i$. In dynamic streams (turnstile models) or situations with prevalent negative weights, where $\sum |f_i| \gg \sum f_i$, sketch-based methods offer significant advantages.
> > >
> > > Our work falls under sketch-based methods. More detailed theoretical comparison between general counter-based methods and sketch-based methods is beyond the scope of this work. Based on your suggestion, we will include a discussion of their relationship with counter-based methods in the related work section.
> > >
> > >
> > >
> > > ## Reply to
> > > >Considering merging I wonder: Suppose that Alice and Bob independently construct Lego sketches of datasets A and B, respectively. How can these sketches be merged?
> > >
> > >
> > >
> > > The mergeability of sketch-based methods has been stated in Agarwal and Cormode's work. In a nutshell, sketch-based methods enable the merging of identically configured sketches through bucket-wise addition, effectively summarizing sketches of multiple datasets in a single sketch. This follows their hash-based additive storage mechanism. For example, in a Count-min sketch, each bucket is updated as: hashed_bucket = hashed_bucket + $f_i$.
> > >
> > >
> > > The merging process of two datasets A and B can be demonstrated by the implementation of the ```Join``` function in the publicly Python library ```Probables```, which provides probabilistic data structures. This function simply sums the corresponding buckets from the two sketches:
> > >
> > >
> > > ```
> > > def join(self, second: "CountMinSketch") -> None:
> > >     ...
> > >     size = self.width * self.depth
> > >     for i in range(size):
> > >         tmp_els = self._bins[i] + second._bins[i]
> > >         ...
> > >         self._bins[i] = tmp_els
> > > ```
> > > Similarly, LegoSketch employs a hash-based additive storage mechanism for memory blocks. As a result, merging two LegoSketch instances corresponding to datasets A and B follows the same bucket-wise summation approach.
> > >
> > > Thank you once again for your timely feedback. If you have any further concerns, please feel free to reach out. We are happy to provide additional clarifications or experiment results.

---

### Official Review · Reviewer_m2XT · 2025-03-13

**Overall Recommendation:** 3

**Summary:**

In this paper, authors have proposed the Lego Sketch, a novel neural sketch designed for data streams. LegoSketch utilize hash embeddings, scalable memory (spread the total space budget across multiple memory blocks, and avoid retraining), memory scanning, and ensemble decoding. During the training phase, the authors have also introduced a self-guided weight loss.
LegoSketch are compared with Count-Min, Count Sketch, Meta-Sketch, D-CMS (Elastic Sketch?) and Learned Augmented Count-Sketch on five real-world datasets and six synthetic data streams (following Zipf distribution but with different skewness). As shown in Figure 6, Lego Sketch has achieved 12% - 80% lower estimation errors. In addition, under distribution shift,  Lego Sketch has demonstrated much better robustness compared to previously learned linear sketch (Meta-Sketch and Learned Augmented Count Sketch).

**Claims And Evidence:**

The methodology is sound and I find most of the claims to be supported.
However, there are a few claims that are problematic.

1) At page 4, the authors claim that "handcrafted sketches face challenges in reconstructing stream global characteristics from the compressed storage". I agree that use sketches to reconstruct global statistic and utilize the statistic to improve query accuracy is challenging. There are such studies. I think these works need to be discussed in the paper.
Please see: Ting, Daniel. "Count-min: Optimal estimation and tight error bounds using empirical error distributions." SIGKDD 2018, and Chen, Peiqing, et al. "Precise error estimation for sketch-based flow measurement." IMC 2021.

2) Based on Theorem 4.3: $P(Err > \epsilon N) < (\epsilon d_{2})^{-1}$. If we denote the failure probability to be $\delta$, then $d_{2}$ is $(\epsilon \delta)^{-1}$. To solve the frequency estimation problem, the total space required for lego sketch is $K$ $d_{1}$ $(\epsilon \delta)^{-1}$ which is larger than the $1/\epsilon log(1/\delta)$ space requirement for Count-Min and Count Sketch. While Lego Sketch has shown good accuracy in practice, from the theory side, it does not offer "superior space-accuracy trade-offs, outperforming existing Sketches" (in abstract).

**Essential References Not Discussed:**

This paper has several foundational work in this space. Please add references to data summaries in the insertion-only model (e.g., MG summary, SpaceSaving, Lossy Counting), and consider to add discussions on the streaming model for LegoSketch (insertion-only, turnstile, bounded deletion).

In section 3.3, the author has mentioned sketches using filtering framework to separate cold and hot items and discussed Elastic Sketch. The author should also consider to add references to related works [1-3].

In addition to Learned Augmented Count Sketch and Meta Sketch, please also add discussion on learning-based frequency estimation [4].

[1] Roy, Pratanu, Arijit Khan, and Gustavo Alonso. "Augmented sketch: Faster and more accurate stream processing." Proceedings of the 2016 International Conference on Management of Data. 2016.
[2] Zhou, Yang, et al. "Cold filter: A meta-framework for faster and more accurate stream processing." Proceedings of the 2018 International Conference on Management of Data. 2018.
[3] Zhao, Fuheng, et al. "Panakos: Chasing the tails for multidimensional data streams." Proceedings of the VLDB Endowment 16.6 (2023): 1291-1304.
[4] Shahout, Rana, and Michael Mitzenmacher. "Learning-based heavy hitters and flow frequency estimation in streams." 2024 IEEE 32nd International Conference on Network Protocols (ICNP). IEEE, 2024.

**Ethical Review Concerns:**

Given AOL dataset has leaked user information (See https://en.wikipedia.org/wiki/AOL_search_log_release), can experimental results based on AOL be included in the paper?

**Ethical Review Flag:**

Flag this paper for an ethics review.

**Ethics Expertise Needed:**

["Privacy and Security"]

**Experimental Designs Or Analyses:**

Is D-CMS (page 6) referring to Elastic Sketch?

**Methods And Evaluation Criteria:**

The evaluation looks good to me. and lego sketch has demonstrate high accuracy in different data domains.

**Other Comments Or Suggestions:**

If D-CMS is Elastic Sketch, then I prefer the authors to use the name proposed by the original author.

**Other Strengths And Weaknesses:**

S1. The methodology of Lego Sketch is sound.

S2. The authors have conducted extensive experiments and demonstrate the strong estimation accuracy of Lego Sketch.

S3. Comparing to other learned sketch, LegoSketch is both robust to distribution shift through hash embedding and avoid diminishing return using self-guided loss.

W1. Some of the claim are not supported. (see `Claims And Evidence`)

W2. Missing important references.

**Questions For Authors:**

1) At the Scalable Memory section, the authors state "using a hash function to uniformly distribute items in streams across these bricks." The uniform distribution is on the cardinality and not on the frequency. Should the memory block size be dependent on the item's aggregated count (i.e., have memory block of varying sizes and use larger block for larger sub-stream)? I can also see the argument for it to be only dependent on cardinality, but I'm curious about the authors opinion.

2) Please also see W1 and W2.

**Relation To Broader Scientific Literature:**

Although some of the techniques are not new, given the strong accuracy advantage of Lego Sketch compared with other learned sketches (Meta Sketch and Learned Augmented Sketch), this work can spark novel research directions in scalable, modular neural architectures for real-time data stream processing.

**Theoretical Claims:**

I didn't not check the proofs in detail, but the theorem 4.1-4.3 are sound.

---

> ### Author Rebuttal · Authors · 2025-03-31
>
> Thank you for your detailed review and constructive feedback! We have addressed your concerns below and are happy to discuss further if needed.
> ## Claims
> ### Reply to
> > ...I agree that use sketches to reconstruct global statistic... There are such studies.....
>
> Thank you. While these methods rely on handcrafted rules to infer error distributions from bucket values, our Scanning Module leverages end-to-end training to directly reconstruct global stream characteristics, such as skewness and cardinality, from bucket values. Intuitively, directly reconstructing global stream characteristics is more effective, as they are the root cause of error distributions.
> That said, we may also explore adapting our end-to-end approach for error distribution reconstruction in the future. We will revise the  descriptions to include discussions of these related methods.
>
> ### Reply to
> > While Lego Sketch has shown good accuracy in practice, from the theory side, it does not offer "superior space-accuracy trade-offs"..
>
> As discussed in Section 4.3 and the Limitations Section (Appendix A), our theoretical bounds are indeed looser than those of handcrafted sketches. However, our key theoretical contribution is establishing the first formal bounds for pure neural sketches, unlike meta-sketches, which don't have theoretical guarantees. Following your suggestion, we will revise the abstract to clarify the "superior space-accuracy trade-offs" is empirically supported.
>
> ## References:
> ### Reply to
> >Add references to insertion-only model and consider to add discussions on the streaming model for LegoSketch.
>
> Lego Sketch, like Meta Sketch, Count-Min Sketch, and Count Sketch, operates in the turnstile model using additive memory buckets, which naturally support arbitrary deletions. We will make this clearer in the revised manuscript. And we will add a dedicated subsection in the Related Work section to discuss insertion-only data summarization techniques (e.g., MG summary, SpaceSaving, Lossy Counting).
>
> ### Reply to
> > ... The author should also consider to add references [1-3].
>
> As noted in Section 3.3 (References in Lines 502 and 510), we have already included references to derivatives like Augmented Sketch and Cold Filter, citing representative works (Zhou et al., 2018; Roy et al., 2016; Hsu et al., 2019; Aamand et al., 2024).
>
> Following your suggestion, we will additionally cite: Zhao, Fuheng, et al. "Panakos: Chasing the tails for multidimensional data streams." This will help readers better understand the technical lineage and connections between these approaches.
>
> ### Reply to
> >Add discussion on Shahout, "Learning-based heavy hitters and flow frequency estimation in streams."
>
> Thank you for bringing this recent work to our attention. The paper employs learning-based methods to separate high/low-frequency items, thereby improving the **insert-only** SpaceSaving algorithm. However, this approach operates in a different streaming model **(insertion-only)** with our proposed sketch method **(turnstile)**. We will include it in the revised Related Work section as an learning-enhanced variant of the insertion-only methods.
>
> ## Other Comments Or Suggestions:
> ### Reply to
> > I prefer the authors to use the original name of Elastic Sketch.
>
> Thank you. We will adopt the original naming convention and refer to them as Elastic Sketch (CMS/Lego) to improve clarity.
>
> ## Questions:
> ### Reply to
>
> >Discussion on whether memory block size can be determined based on frequency or cardinality.
>
> We appreciate the insightful question. Our current design of Lego Sketch employs uniform item distribution across fixed-size memory blocks based on cardinality. As formalized in Theorem 4.2, this ensures that the skewness distribution within each sub-block closely matches the global stream skewness. Consequently, all blocks exhibit similar statistical properties (skewness, cardinality, and memory allocation) - a desirable feature for scalable memory design, as it ensures stable statistical characteristics during scaling memory.
>
> Indeed, memory block size does not have to be determined solely by cardinality.
> A frequency-aware approach could enhance efficiency by assigning larger blocks to high-frequency substreams but may introduce risks under distributional shifts. Although such strategies have potential benefits, they fall outside the scope of our current work. We would be glad to explore this direction in future — for example, using learning-based predictors to separate high- and low-frequency items and adjust block sizes accordingly.
>
> ## Ethical Concerns：
> ### Reply to
> >  AOL dataset has leaked user information...
>
> Thank you for pointing this out!
> This dataset has been widely used in prior studies, so we followed this practice without realizing the issues. We will remove the results derived from this dataset. The remaining four real-world datasets and six synthetic datasets would be sufficient to demonstrate the superior performance of our work.

---

### Official Review · Reviewer_EjyL · 2025-03-13

**Overall Recommendation:** 3

**Summary:**

This paper proposes a method for estimating the frequency of items in a data stream by means of sketching of embedding vectors of the items. It claims scalable memory use by means of multiple "bricks". It compares with hand-crafted and neural sketch methods on a number of datasets.

**Claims And Evidence:**

The proposed method is found to have consistently lower error metrics as a function of memory budget relative to the competing methods, on a number of datasets. It is also found to have superior robustness to distribution shift on synthetic datasets.

**Essential References Not Discussed:**

As I am not an expert on hashing, I cannot say if essential references are missing.

**Experimental Designs Or Analyses:**

I am not expert in this area, but all experiments seem sound. I do not find any issues.

**Methods And Evaluation Criteria:**

I am not familiar with the datasets in this area, but the metrics are simple and reasonable.

**Other Comments Or Suggestions:**

None.

**Other Strengths And Weaknesses:**

The text is very well written and easy to follow.

**Questions For Authors:**

None.

**Relation To Broader Scientific Literature:**

I am not an expert on sketching. Thus, I cannot say much on the novelty of the approach, the benchmarks, the competition and the state of the art.

However, simply by reading the method, it is hard to see that this is a novel method. Every component is very simple. For example, the authors claim to introduce a novel "normalized multi-hash embedding". But this is just some standard embedding followed by some standard hashing followed by normalization. It is extremely difficult to imagine that this is novel. Same goes for all components.

Despite not seeing any novel idea, I rate the paper with weak accept, simply because of my lack of expertise on the subject.

## Post-Rebuttal

Given my lack of familiarity with the subject and the positive scores of the other reviewers, I am keeping my score.

However, on my concern on novelty, I strongly advise the authors to revise their manuscript according to the post-rebuttal discussion and according to their commitments: That they will specify the technical novelty of each component and of the entire approach relative to the state of the art. Not just qualitative properties of the methods, but in terms of ideas, methods and algorithms. For example, "Scaling" an existing idea is not really new.

**Theoretical Claims:**

The proof of Theorem 4.1 appears correct.

---

> ### Author Rebuttal · Authors · 2025-03-31
>
> Thank you for your feedback! We have provided responses below and would be happy to engage in further discussion if needed.
>
> Neural sketch represents a promising new direction in the sketch field long dominated by handcrafted designs. Existing neural sketches such as Meta Sketch demonstrate feasibility but suffer from limited scalability, insufficient accuracy, and poor deployability. Lego Sketch takes the critical next step—by introducing architectural and training innovations that make neural sketch accurate, scalable, and practically usable. This transforms neural sketch from a conceptual idea into a deployable core technique for data stream processing.
>
> More specifically, both handcrafted and neural sketch structures are designed to prioritize simplicity and effectiveness, using minimal memory to cope with infinite data streams. For over a decade, even the latest handcrafted sketches have primarily relied on a few hash functions and a two-dimensional array with straightforward decoding rules. Similarly, leading neural sketches like Meta Sketch rely on basic architectures, using simple MLPs across modules. This highlights the core challenge in sketching: achieving practical scalability with limited space and high accuracy.
>
> Lego Sketch introduces significant advancements in this regard, focusing on the emerging domain of neural core sketching. Unlike Meta Sketch, which suffers from limited scalability and frequent retraining needs, Lego Sketch incorporates scalable embedding and memory mechanisms, ensuring seamless deployment without retraining. This fundamental contribution directly addresses a major limitation in the field.
>
> Moreover, Lego Sketch demonstrates technical novelty through its structural design. Modules such as the Ensemble Decoding Scanning Module and the self-guided loss contribute to accuracy improvements, as evidenced by comprehensive ablation studies. Experimental results further validate its competitive throughput, underscoring its balanced performance. By refining all components beyond Meta Sketch, Lego Sketch advances the state of the art in neural sketching.

---

> > ### Comment · Reviewer_EjyL · 2025-04-06
> >
> > I thank the authors for the feedback.
> >
> > Given my lack of familiarity with the subject and the positive scores of the other reviewers, I believe I will keep my score.
> >
> > However, on my concern on novelty, I have to say that the authors' response is purely on qualitative properties of the methods. What is needed is to understand novelty in technical terms, in terms of ideas, methods and algorithms. "Scaling" an existing idea is not really new. Such discussion should be in the paper, not just the rebuttal.

---

> > > ### Author Response · Authors · 2025-04-07
> > >
> > > Thank you for your feedback!
> > >
> > > We will provide a clear explanation highlighting the novelty of each component in the Methodology Section of revised manuscript.
> > >
> > > Specifically:
> > >
> > > - The **Scalable Embedding** leverages a normalized multi-hash technique to address, for the first time, the cross-domain generalization challenges of Neural Sketch, with theoretical analysis in Section 4.1.
> > > - The **Scalable Memory** dynamically expands memory capacity via memory block stacking, overcoming previous scalability bottlenecks while ensuring prediction accuracy, with analysis in Section 4.2.
> > > - The **Scanning Module** proposes an entirely new approach to reconstructing stream characteristics via end-to-end training, aiding the decoding process and validated through ablation studies.
> > > - The **Ensemble Decoding** achieves more accurate estimation and provides the first theoretical error guarantee for Neural Sketch, as described in Section 4.3.
> > > - The **Self-guided Weighting Loss** employs self-supervised weighting to effectively balance Lego Sketch training across diverse data streams, addressing degradation issues in prior Neural Sketch under certain streams.
> > >
> > > We will also highlight the novelty of each module more prominently in both the Introduction and Conclusion sections.

---

### Official Review · Reviewer_MS4A · 2025-03-17

**Overall Recommendation:** 3

**Summary:**

This paper introduces the Lego Sketch, a neural network-augmented sketch for frequency estimation. The sketch consists of several learnable components: a variant of the hash embedding layer, a "memory scanning" module that estimates global characteristics of the stream, and an "ensemble decoding" module that returns an estimated item frequency using a decoder network. The design of the Lego Sketch allows for the size of the memory data structure to be modified without retraining the learnable components of the sketch. The empirical evaluation compares the proposed sketch against several classical and learned baselines, demonstrating improved frequency estimation at lower memory budgets.

**Claims And Evidence:**

The main claim in the paper is that the proposed sketch improves on previously proposed approaches for the frequency estimation task. This claim is supported by the empirical evaluation, which shows that the Lego Sketch improves on both "handcrafted" sketches and the neural network-augmented Meta-sketch in both absolute and relative error metrics.

**Essential References Not Discussed:**

-

**Experimental Designs Or Analyses:**

-

**Methods And Evaluation Criteria:**

The proposed methods and evaluation criteria are reasonable.

**Other Comments Or Suggestions:**

- The legibility of the figures in Sec. 5 should be improved. With the current scaling of the y-axis, it is difficult to distinguish between the accuracy curves in several subplots.

**Other Strengths And Weaknesses:**

-

**Questions For Authors:**

1. At lower memory budgets, the memory overhead of the sketch's neural network components becomes a larger fraction of overall memory usage. This is of particular concern when deploying the sketch on resource constrained edge devices. How much memory do the Lego Sketch's NN components use, and is the Lego Sketch still competitive with the handcrafted baselines when this memory overhead is included?
2. Given the goal of reducing memory usage, it is natural to consider using lower precision numerical representations. What is the numerical precision used in the experiments reported in the submission?
3. Following up on the previous question, how well does the sketch perform when its NN components are quantized to lower precision to reduce memory consumption (e.g., to 4 bits per parameter)? How well does the sketch perform when the buckets of the "scalable memory" are quantized to lower precision?

**Relation To Broader Scientific Literature:**

This paper is related to the literature on memory efficient sketching algorithms for frequency estimation in data streams, and to more recent work on neural network-augmented sketching methods.

**Theoretical Claims:**

I did not check the correctness of the paper's theoretical claims.

---

> ### Author Rebuttal · Authors · 2025-03-31
>
> Thank you for your valuable feedback! Below, we address your concerns in detail.
>
> ## Other Comments Or Suggestions:
>
> ### Reply to
> > The legibility of the figures in Sec. 5 should be improved.
>
>
>
> Thank you for your suggestion. We will improve the readability of the figures accordingly.
>
> ## Questions For Authors:
>
> ### Reply to
> >Question 1： How much memory do the Lego Sketch's NN components use, and is the Lego Sketch still competitive with the handcrafted baselines when this memory overhead is included?
>
> The NN components of Lego Sketch are highly space-efficient, containing only around 5K parameters, which equates to a storage size of 20KB with 32-bit precision. Given that the total memory usage in our experiments ranges from 0.3MB to 140MB, the NN components contribute only a negligible fraction, i.e., between 6.7% and 0.014% of the total memory. Even when accounting this overhead, Lego Sketch remains highly competitive and significantly outperforms handcrafted baselines. For example, on the LKML dataset, with a total usage of 620KB (600KB for storage plus 20KB for NN components), Lego Sketch records an AAE of only 2.55, which is lower than CM (2.58) and C Sketch (2.95), even when these baselines use 1100KB of memory.
>
> |   LKML Dataset, AAE    | 600KB  | 1100KB | 1600KB | 2100KB | 2600KB | 3100KB | 3600KB |
> |------|------|------|------|------|------|------|------|
> | Lego Sketch(+20KB) | 2.55 | 1.67 | 1.03 | 0.61 | 0.3  | 0.18 | 0.11 |
> | CM  Sketch | 6.75 | 2.58 | 1.33 | 0.78 | 0.5  | 0.35 | 0.25 |
> | C  Sketch  | 5.36 | 2.95 | 1.95 | 1.41 | 1.08 | 0.85 | 0.69 |
>
> Generally, the NN overhead for neural sketches is typically not accounted for, similar to other neural network-augmented methods, like learned sketches and Meta sketch. This is because this overhead can be further amortized across multiple deployed instances.
>
> ### Reply to
> >Question2：What is the numerical precision used in the experiments reported in the submission?
>
> Lego Sketch uses the default 32-bit precision, and with only 5K parameters, the memory overhead is negligible.
>
> ### Reply to
> > Question3：how well does the sketch perform when its NN components are quantized to lower precision to reduce memory consumption (e.g., to 4 bits per parameter)? How well does the sketch perform when the buckets of the "scalable memory" are quantized to lower precision?
>
>
> Neural sketch's memory module requires frequent updates, which will frequently triggers quantization and dequantization operations when updating the quantized memory. This significantly impacts throughput. Future research could focus on designing efficient quantization algorithms tailored for high-update scenarios.
>
> For the NN components, this could be feasible. However, Lego Sketch has only 5K parameters, which is significantly small—consuming just 0.014% to 6.7% of the total memory storage in experiments. Compared to Meta Sketch, another neural sketch with 52K parameters, Lego Sketch is smaller by an entire order of magnitude. Thus, further quantization of the NN components provides minimal benefit.
>
> Following your suggestion, we attempted reducing precision from 32-bit to 16-bit, saving 10KB in storage of NN components. However, under small memory budgets, the error notably increased—e.g., under 600KB, Lego Sketch's error rose from 2.55 to 6.23. In contrast, for larger memory budgets, such as above 2600KB, the error remained consistent within two decimal places.
>
>
> |  LKML Dataset, AAE    | 600KB  | 1100KB | 1600KB | 2100KB | 2600KB | 3100KB | 3600KB |
> |------|------|------|------|------|------|------|------|
> | Lego Sketch(+20KB, 32bits) | 2.55 | 1.67 | 1.03 | 0.61 | 0.3  | 0.18 | 0.11 |
> | Lego Sketch(+10KB, 16 bits) | 6.23 | 2.29 | 1.05 | 0.61 | 0.3  | 0.18 | 0.11 |

---

### Decision · Program_Chairs · 2025-05-01

**Decision:**

Accept (poster)

**Comment:**

The paper proposes a modular, memory-augmented neural network architecture for frequency estimation on data streams. There were some concerns raised about the novelty of the proposed approach, and comparison to work in the insertion-only model such as Misra-Gries. On further discussion, the reviewers were generally comfortable with the comparison baselines, provided some more discussion is added to the paper. Regarding novelty, though the paper uses existing components, it achieves good results in practice, and the resulting modular framework could be useful for future work. Therefore, I recommend the paper to be accepted.